# Soil-Transmitted Helminths and Anaemia: A Neglected Association Outside the Tropics

**DOI:** 10.3390/microorganisms10051027

**Published:** 2022-05-13

**Authors:** Sara Caldrer, Tamara Ursini, Beatrice Santucci, Leonardo Motta, Andrea Angheben

**Affiliations:** Department of Infectious—Tropical Diseases and Microbiology, IRCCS Sacro Cuore Don Calabria Hospital, Negrar di Valpolicella, 37024 Verona, Italy; sara.caldrer@sacrocuore.it (S.C.); beatrice.santucci@sacrocuore.it (B.S.); leonardo.motta@sacrocuore.it (L.M.); andrea.angheben@sacrocuore.it (A.A.)

**Keywords:** anaemia, soil-transmitted helminths, hookworms, whipworm, roundworm, threadworm, non-endemic countries

## Abstract

Anaemia is an important cause of morbidity and mortality globally. Among infectious agents responsible for anaemia, helminthic infections are often neglected, particularly in non-endemic countries. However, they should not be neglected in this setting, as international travel and migration are on the rise. In this narrative review, we aimed to describe soil-transmitted helminths as a cause of or contributing factor to anaemia, focusing on hookworms (*Necator americanus* and *Ancylostoma duodenale*), the whipworm (*Trichuris trichiura*), the roundworm (*Ascaris lumbricoides*), and the threadworm (*Strongyloides stercoralis*). A general review on the epidemiology, lifecycle, and clinical spectrum of anaemia is proposed, with a special focus on helminthic infections’ association with anaemia as well as the diagnostic approach, which are both particularly important in non-endemic settings.

## 1. Introduction

Anaemia, defined as having a haemoglobin (Hb) concentration lower than normal for certain sex, age, and ethnicity, is a serious global public health problem, affecting high-, middle-, and low-income countries (HICs, MICs, and LICs, respectively) [1]. The World Health Organization (WHO) defines anaemia as an Hb level below 13 g/dL for males, 12 g/dL for nonpregnant women, and 11 g/dL for pregnant women; for children aged 6 months to 2 years, anaemia refers to Hb concentrations less than 11 g/dL, whilst for those aged 2 years to 12 years, anaemia refers to Hb concentrations less than 12 g/dL. The prevalence of anaemia varies significantly across regions and countries [1]. In HICs, the prevalence is estimated at 9%; conversely, the prevalence is 43% in low- and middle-income countries [2]. Anaemia results from a number of causes, including iron deficiency, micronutrient deficiencies, inherited or acquired disorders of haemoglobin synthesis, red blood cell production or survival, and acute and chronic infections [3].

The stark contrast in the prevalence of anaemia between HICs and low-middle-income countries (LMICs) is partially due to different levels of exposure to various determinants of this condition, including infectious agents [2]. However, globalization, increased mobility of people, and international migration are important factors that can change the epidemiological pattern of infectious diseases, including those contributing to anaemia [4]. 

A wide range of infectious agents may at some point be associated with chronic anaemia, including bacteria (tuberculosis), viruses (human immunodeficiency virus), protozoa (leishmaniasis), and helminths (soil-transmitted helminths and schistosomiasis) [5].

Soil-transmitted helminth (STH) infections are among the most common human parasitic infections worldwide and affect the poorest and most deprived populations, with limited access to adequate water, sanitation, and hygiene [6]. Although most prevalent in LMICs, STH infections also occur in vulnerable populations in HICs [7]. Different studies conducted in HICs have reported a high prevalence of helminthic infections among migrants and refugees [8].

The main species that infect humans are hookworms (*N. americanus* and *Ancylostoma duodenale*), the whipworm (*Trichuris trichiura*), the roundworm (*Ascaris lumbricoides*), and the threadworm (*Strongyloides stercoralis*). In 2010, the WHO estimated that more than 800 million children needed treatment for STH infections [6]. This estimate does not include *Strongyloides stercoralis*, which infects around 600 million people globally, according to recent projections [9].

In this review, we focused on STHs as a cause of anaemia, highlighting key factors such as pathogenesis, diagnosis, and management, as they can be a particular challenge for physicians in non-endemic countries. Indeed, with increased international travel and migration of populations, health practitioners can expect to encounter STH infections in their patients presenting with anaemia.

## 2. Methods

We searched PubMed and Embase for articles published up until 31 January 2022. The search terms we used were “anaemia”, “helminth”, “hookworms”, “whipworm”, “roundworm”, “threadworm”, “Ancylostoma”, “Necator”, “Trichiuris”, “Ascaris”, and “Strongyloides”. In our search, we included articles published in English, French, Spanish, Portuguese, and Italian.

## 3. Global Burden

Hookworms (*A. duodenale* and *Necator americanus*) infect an estimated 472 million people worldwide [10]. *Necator americanus* accounts for the majority of human–hookworm cases worldwide; it is particularly widespread in the Americas, most of Africa, Southern China, and Southeast Asia. *Ancylostoma duodenale* is commonly endemic in North Africa, in the Mediterranean region, and in northern regions of India and China. In some parts of Africa, China, India, and elsewhere, mixed human infections with *N. americanus* and *A. duodenale* are not uncommon [11]. Hookworm prevalence and infection intensity are highest in adults, although children are often infected too [11]. Hookworms are the most significant human parasitic infection, causing the loss of 1.8 million disability-adjusted life years (DALYs) worldwide in 2001 [12]. Anaemia was included among the range of sequelae considered [12].

*Trichuris trichiura*, also called the whipworm for its shape that resembles a whip, is a parasitic roundworm affecting an estimated 600–800 million people worldwide. *Ascaris lumbricoides* infects an estimated 804 million people, most commonly children and adolescents [10]. *Trichuris trichiura* and *A. lumbricoides* carry a major burden of disease, causing the loss of around 1.6 to 6.4 million DALYs [12]. Mostly, people who are affected live in areas with poor hygiene and sanitation and in humid climates of the tropical and subtropical regions. School-age children harbour the highest adult–worm burden [13].

Strongyloidiasis is a disease that develops in people infected with *S. stercoralis* (rarely, a related species, *S. fülleborni*, can affect humans), occurring widely in tropical and subtropical areas, and in regions with temperate climates. According to the WHO, an estimated 30–100 million people are infected worldwide, but this picture is probably largely underestimated. Precise data on the prevalence of this infection are unknown in most endemic countries. In a recent paper, the estimated global prevalence of strongyloidiasis in 2017 was 8.1%, corresponding to around 614 million people infected worldwide [9]. Endemic areas are not fully known, but the parasite has been recorded in tropical and subtropical regions and in temperate climates. Due to the chronic nature and life-long persistence in the host of strongyloidiasis, its burden persists several decades after a person has left a disease-endemic area. No public health strategies for controlling the disease are currently active at a global level [6], but the WHO now recommends a control program for strongyloidiasis; controlling this disease is one of the targets of the WHO Roadmap to 2030 [14].

Despite the fact that there is no information on the prevalence of STH infections in non-endemic countries, awareness in the field has increased in recent years due to the evidence of outbreaks of infections in HIC. A survey performed in Alabama found that 34% of the participants tested positive for *N. americanus* by molecular technique. Although it was thought that hookworm infections in the United States had been eradicated for decades, one in three people in a poor area of Alabama tested positive [15].

Moreover, very few documents are available in the literature regarding STH infections in travellers. An important example of these documents is the case report of a double hookworm and *S. stercoralis* infection in a young European tourist returning from Southeast Asia. In this case, the suspicion of parasitic infection was due to diarrhoea presented after returning from endemic areas as well as elevated levels of eosinophils [16].

These observations suggest that although helminth infections are often overlooked in non-endemic countries, there are clear concerns that should not be neglected in this context, as international travel and migration are on the rise.

## 4. Biological Features and Lifecycle

Hookworms are adult nematode worms with a buccal capsule armed with teeth (*A. duodenale)* or cutting plates (*N. americanus*). After the female worm releases thousands of eggs each day with stool in warm soil, the eggs hatch after 5–10 days and the larvae begin their maturation process. Rhabditiform larvae (L1 and L2) become infectious after moulting into L3 larvae, which can survive several weeks in the soil [7]. Whereas *N. americanus* larvae infect humans by transdermal or percutaneous transmission, *A. duodenale* larvae can both penetrate the skin or infect orally. Furthermore, *A. duodenale* can pass into breast milk (transmammary transmission) and even cross the placenta (transplacental transmission). Finally, after maturation to fourth-stage larvae (L4), hookworms become able to migrate and live in host tissues; to pass through the pulmonary capillaries, where they are transported and penetrated into the alveolar wall (causing pneumonia); and to pass on to the larynx, where they may be ingested. The larvae mutate into mature worms in the small intestine in 1–2 months and can survive for months (*A. duodenale*) or even years (*N. americanus*).

*Trichuris trichiura* and *A. lumbricoides* infect humans through faecal–oral transmission. A human host consumes infectious eggs deposited in moist soil via food or dirty hands. Once the embryonated eggs are ingested, the larvae attach to the intestinal villi and mature into adult worms. *Ascaris lumbricoides*, but not *T. trichiura,* may migrate to the pulmonary vasculature, lodge in capillaries, and rupture into the alveolar space. Female worms lay thousands of eggs daily for several years. The eggs pass in the stool and embryonate in warm, moist soil, where they can survive for months [7]. 

*Strongyloides stercoralis* transmission occurs when the skin of the host is exposed directly to contaminated soil; infective (filariform–L3) larvae penetrate the skin, pass through the lymph to the venous bloodstream, and reach the lungs. In the alveoli, larvae pass into the airways and are finally ingested into the digestive tract. In the small intestine, they mature into female adults and produce eggs [17]. Eggs usually hatch in the gut lumen, and rhabditiform (L1) larvae develop and are passed in the stool, potentially starting a new extra-host cycle. Some rhabditiform larvae eventually become filariform in the large intestine, where they are able to pass the perianal skin or the terminal intestine wall perpetuating an auto-infective cycle without the extra-host life cycle. Infective larvae, released with faeces under defined humidity and temperature conditions of the environment, mature into male and female adults, which can begin a sexual cycle leading to the production of eggs and new L3 larvae. From a clinical point of view, the unique auto-infective cycle of *S. stercoralis* is a key element determining the chronic nature of the disease and the risk of highly lethal disseminated strongyloidiasis in cases of immune-suppression, which accelerates the auto-infective cycle. Walking barefoot is considered a major risk factor for acquiring the infection [17].

## 5. Pathophysiology and Clinical Spectrum

Helminth infections generally induce a predominantly T-helper-2 cell (Th2) response. The immunology of hookworm infections has received less attention than other human–helminth infections, primarily due to the difficulty of maintaining the hookworm life cycle in an animal model [18]. During acute infection, elevated IgE titres, interleukin 5, and eosinophilia are frequent. The inflammatory response, characterized by eosinophilia and mastocytosis, is directed against the adult hookworm or L3 migrating infectious larvae [19]. The penetration of hookworm larvae into the skin causes intense itching, so-called ‘ground itch’, and often a cutaneous rash. Eosinophilic pneumonia with cough, dyspnoea, and haemoptysis might occur during larval pulmonary migration (i.e., Loeffler syndrome). Wakana disease, characterized by nausea, fever, vomiting, cough, shortness of breath, and hoarseness, could be caused by direct oral infection of *A. duodenale.* Additionally, *A. duodenale* and *N. americanus* larvae might cause a type-1 hypersensitivity reaction during pulmonary migration (Loeffler syndrome). Once worms are established in the small intestine, adult worms burrow their teeth into the mucosa, causing blood loss [18,20]. The clinical picture comprises asthenia, abdominal pain, and diarrhoea, and infections can lead to pallor, tachycardia, tachypnoea, oedema, abdominal tenderness, occult faecal blood, and occasionally melaena. In heavy-intensity infections, blood loss can result in severe anaemia. Individuals prone to under-nutrition and malaria, such as children and pregnant women, are especially vulnerable [18,20].

Similar to other helminthiases, *T. trichiura* and *A. lumbricoides* induce a predominant Th2 immune response and eosinophilia. Trichuriasis is also usually asymptomatic; otherwise, people with heavy infections can experience chronic abdominal pain as well as the painful passage of stool that contains a mixture of mucus, water, and blood. Rectal prolapse can also occur. Children may develop anaemia, growth retardation, and even impaired cognitive development, probably due to iron deficiency and poor nutrition secondary to worm burden [7]. *Ascaris lumbricoides* can trigger a type-1 hypersensitivity reaction to larvae (i.e., Loeffler syndrome), and adult worms may induce intestinal disease. The majority of *A. lumbricoides* infections cause mild or no symptoms in endemic areas. The clinical manifestations are related to worm burden, although the association is not linear, including upper gastrointestinal bleeding, small bowel obstruction, volvulus, intussusception, peritonitis, intestinal ischemia, and perforation [21].

Like other geo-helminthiases, the risk of *Strongyloides* infection is associated with a lack of hygiene and sanitation, thus making children especially vulnerable [21]. The majority of infections occur in the preschool-age group, and symptoms of acute strongyloidiasis go unnoticed or become attributed to other diseases such as acute respiratory infections, asthma, or paediatric rashes [22]. Löffler syndrome represents the symptomatic expression of the acute stage of the disease. Then, in the chronic phase, strongyloidiasis may cause gastrointestinal symptoms such as intermittent vomiting, abdominal pain, diarrhoea, constipation, and borborygmus. Moreover, pulmonary symptoms such as cough, wheezing, recurrent bronchitis, or dermatologic manifestations (pruritus and urticaria) may occur as a consequence of larval auto-infective cycle and immune reaction to the helminths [23]. However, it has been estimated that half of the cases are asymptomatic. Although strongyloidiasis is usually a mild illness, the infection may become severe and life-threatening in cases of immunodeficiency when the auto-infective cycle accelerates without control and larvae invade the bloodstream in large amounts and/or reach virtually all organs, including the central nervous system [24].

## 6. Association with Anaemia

The mechanisms by which hookworms induce blood loss are multi-factorial and are summarized in Table 1. The severity of chronic blood loss depends on the intensity of infections, the species of hookworm, host iron reserves, and other factors such as age and co-morbidity [2].

Most instances of hookworm-induced blood loss are results of gut–epithelial and endothelial barrier disruptions at the entry site [18]. Hookworms use their teeth (*A. duodenale*) or cutting plates (*N. americanus*) to attach to the mucosa and submucosa. They also secrete parasite-derived anticoagulants that prevent blood from clotting and that negatively impact the host’s inflammatory response [18,23,24]. Furthermore, adult worms’ active feeding, which partially digests the host’s erythrocytes, causes significant and direct blood loss [18]. *Ancylostoma duodenale* is thought to be a wasteful feeder (not all the blood it ingests is digested) and is responsible for blood loss that is as much as ten times greater than that caused by *N. americanus* [25].

In detail, adult hookworms of *N. americanus* and of *A. duodenale* are thought to ingest 0.05 to 0.2 mL of blood per day per worm, with an average daily blood loss of 26.4 mL in adult subjects infected with *A. duodenale* [26]. A heavy *N. americanus* infection can produce daily losses of >1 mL of blood [18]. The intensity of hookworm infection is generally classified in eggs per gram of faeces (EPG). The WHO categorizes hookworm infections as light (≤1999 EPG), moderate (2000–3999 EPG), and heavy (≥4000 EPG). Hookworm disease is defined as the presence of moderate-to-heavy hookworm infections, and when blood loss exceeds the host’s intake and reserves of iron and proteins, chronic iron-deficiency anaemia (IDA) and the accompanying hypoalbuminaemia and hypoproteinaemia occur. Hypoproteinaemia can cause peripheral oedema; ascites; and in severe cases, a clinical picture that resembles kwashiorkor [2,18].

Anaemia following trichiuriasis is principally a consequence of chronic blood oozing in the caecum and colon through the mucosal entry sites of the adult worms. Blood loss due to infection with *T. trichiura* has been estimated to be around 0.005 mL a day per worm [27,28]. Therefore, the contribution to anaemia is more frequent in cases of heavy infections (defined as more than 800 worms or the release of more than 5000 eggs per stool gram). Trichiuriasis is often a co-infection. Therefore, it is difficult to differentiate its contribution to anaemia and other clinical manifestations from that of other helminths, such as hookworms or ascariasis [7]. Anaemia can be severe in vulnerable groups, such as pregnant women, whose iron reserves may be severely depleted [29]. Heavy *T. trichiura* infection in children is a severe illness associated with iron-deficiency anaemia, growth deficiency, abdominal discomfort, chronic mucoid diarrhoea, rectal bleeding, rectal prolapse (a consequence of increased straining and/or peristalsis), and finger clubbing [30]. Together with chronic mucosal bleeding, inflammation caused by *T. trichiura* may affect the nutritional status of children through impaired nutrient absorption and disruption of intestinal flora, and one can argue that this can contribute to growth retardation and anaemia [13]. Not only children but also adults not reached by mass drug administration programs can present severe iron deficiency and anaemia as a consequence of trichiuriasis [31]. 

The occurrence of anaemia following ascariasis is mainly due to intestinal obstruction and ileum perforation [32]. Moreover, chronic ascariasis may be associated with malnutrition, as a direct effect of inadequate nutrients absorption [33].

Few studies in the literature have addressed the issue of anaemia and strongyloidiasis, and the results are often presented as disaggregated. Therefore, strongyloidiasis health effects in the context of polyparasitism cannot be evaluated per se. A systematic review and meta-analysis on morbidity associated with chronic strongyloidiasis cannot evaluate anaemia as a reported sign [34]. However, a review regarding the effects of strongyloidiasis among pregnant women and children is available [35]. The authors concluded that current evidence does not support strongyloidiasis as a cause of anaemia in pregnant women or children [36]. Since almost all surveys studied *Strongyloides stercoralis* as a part of polyparasitism, one can argue that its prevalence and pathogenicity could be underestimated. Nevertheless, strongyloidiasis is common in under-nourished children and among poorer populations, and its presence is linked to low weight at birth and wasting. Anaemia is commonly accompanying these conditions [35].

Impaired nutrient absorption and disruption of the intestinal flora can contribute to growth retardation and anaemia in children and in pregnant women. In the chronic phase, strongyloidiasis causes intermittent symptoms that mostly affect the intestine, such as abdominal pain, and intermittent or persistent diarrhoea [35].

**Table 1 microorganisms-10-01027-t001:** Mechanisms by which soil-transmitted helminths cause, or may contribute to, anaemia, and the association between the disease severity and worm burden.

	Hookworms		Whipworm	Roundworm	Threadworm
	** *Ancylostoma duodenale* **	** *Necator americanus* **	** *Trichiuris trichiura* **	** *Ascaris lumbricoides* **	** *Strongyloides stercoralis* **
	**Direct association with anemia**		**Indirect association with anemia**		
	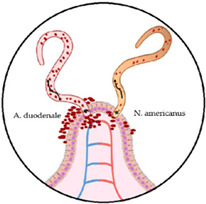		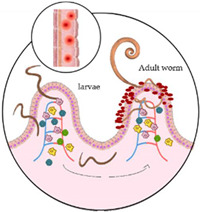	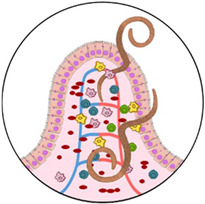	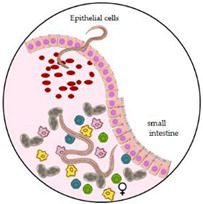
Mechanism/s	Attachment to the gut mucosa and submucosa through their teeth	Attachment to the gut mucosa and submucosa through their cutting plates	Gut epithelial and endothelial barrier disruption in the entry site	Gut epithelial and endothelial barrier disruption in the entry site	Gut epithelial and endothelial barrier disruption in the entry site
	Secretion of anticoagulants to prevent blood clotting. Downregulation of the host inflammatory response	Secretion of anticoagulants to prevent blood clotting and downregulate the host inflammatory response	Rupture of capillaries and arterioles within the intestinal tissue, leading to blood loss	Rupture of capillaries and arterioles within the intestinal tissue, leading to blood loss	Probably poor nutrient absorption and disruption of intestinal flora
Severity	0.2 mL/day/worm (standard deviation of ±0.045 mL) [25]. Blood loss 10-fold higher than *N.americanus*	0.05–0.15 mL/day/worm -heavy infections can produce losses of >1 mL daily [18]	0.005 mL/day/worm [29]	Not determinable	Not determinable
	Host’s erythrocytes are digested in the gastrointestinal tract of the hookworm; depletion of iron reserves, hyproteinaemia	Host’s erythrocytes are partially digested by the gastrointestinal tract of the hookworm; depletion of iron reserves, hyproteinaemia	Chronic blood oozing in the caecum and colon through the mucosal entry sites; depletion of iron reserves	Intestinal obstruction and perforation; malnutrition	Not determinable
	In heavy-intensity infections, blood loss can result in severe anaemia	The severity of blood loss positively correlates with parasite burden	Severe anemia in vulnerable groups (e.g., pregnant women and children)	Severe anemia in vulnerable groups (e.g., pregnant women and children)	Not determinable
Accompanying clinical elements	In moderate/heavy infections, IDA, hypoalbuminaemia and hypoproteinaemia can occur; hypoproteinaemia can cause peripheral oedema, ascites, and, in severe cases, a clinical picture that resembles kwashiorkor	In moderate/heavy infections, IDA, hypoalbuminaemia and hypoproteinaemia can occur; hypoproteinaemia can cause peripheral oedema, ascites, and, in severe cases, a clinical picture that resembles kwashiorkor. In heavy infections, anemia ensues even when adequate dietary intake is maintained	IDA and poor nutrient adsorbition. Impaired nutrient absorption and disruption of intestinal flora can contribute to growth retardation and anemia in children. Massive infantile trichiuriasis is a severe illness including IDA, chronic mucoid diarrhoea, rectal bleeding, rectal prolapse	Gastrointestinal bleeding, small bowel obstruction, volvulus, intussusception, peritonitis, intestinal ischemia, and perforation. Chronic ascariasis may be associated with malnutrition, which is caused by malabsorption of dietary protein, fat, vitamins.	In the chronic phase, strongyloidiasis may cause intermittent symptoms that mostly affect the intestine (abdominal pain and intermittent or persistent diarrhoea)

## 7. Diagnostic Approach

Diagnosis of anaemia caused by hookworm infection is challenging for physicians in temperate climates as they may be less familiar with these parasites than physicians in tropical regions of the world. Diagnosis requires a high level of suspicion, based on knowledge of the parasites’ geographical distributions; thus, the patient’s history is a key element in making a comprehensive differential diagnosis [36,37]. Moreover, clinical examination may help to narrow the diagnosis, including signs and symptoms of STH infections other than anaemia (see the Pathophysiology and Clinical Spectrum section). Measurement of the haemoglobin concentration is the first step for diagnostic purposes. 

Moreover, the determination of the degree of anaemia, typically featuring microcytic and hypochromic erythrocytes in blood smear analysis, is important/essential in all hookworm infections. An analysis of blood smears from patients who are heavily infected (often) shows the presence of microcytic hypochromic red blood cells [38,39]. Complete blood counts often demonstrate the presence of eosinophilia, a classic hallmark of nematode infection. Elevated eosinophil levels cause gastrointestinal (diarrhoea) and neurological symptoms (such as blurred vision and slurred speech) and should be considered especially in travellers returning from endemic areas that are commonly prone to acute, light-intensity infections. Individuals living in, and emigrants from, endemic areas are repeatedly exposed to hookworms, some with very high worm burdens and commonly presenting chronic diseases [37,38]. In those individuals, other causes of anaemia, such as malaria, HIV/AIDS, and haemoglobinopathies may be excluded.

Severe hookworm anaemia clinically resembles IDA, with physical signs and symptoms as described previously [39,40].

Hookworms might exacerbate anaemia in patients with underlying nutritional deficiencies. Hookworm infection is diagnosed by identifying hookworm eggs from faeces under light microscopy. However, hookworm eggs are not detected during the period in which the larvae make their way through the vascular system to the small bowel, and this can take more than 5 weeks [39]. In travellers returning from endemic areas, furthermore, parasite eggs might not appear in stool for months after exposure or symptom onset. Egg production does not necessarily occur continually, as it can be affected by the nutritional status of the host. Methods to determine egg concentrations are limited by day-to-day variability and by uneven distribution of eggs in stool. This might lead to false-negative results, especially in low-intensity infection and in analysis of post-treatment cases. It is advisable to examine more than one stool sample collected over different days for increasing the sensitivity of the microscopy analysis [39,40].

Polymerase Chain Reaction (PCR) assays are being developed both for clinical and for public health management. Indeed, one such multiplex quantitative PCR faecal assay can potentially even differentiate hookworm species [40,41]. However, these kind of tests are not yet broadly available. Although rarely used, capsule endoscopy can identify hookworms [42]. 

The diagnosis of trichiuriasis and ascariasis broadly relies on clinical presentation, on the detection of eggs (microscopy-based diagnostic methods), or on parasite DNA in the faeces (molecular-based methods) [13]. Molecular-based assays are currently being developed and used, aiming to improve sensitivity and specificity [42]. Upper endoscopy might support the diagnosis of duodenal ascariasis; endoscopic retrograde cholangiopancreatography can be used to remove worms from ducts or duodenum [43].

Colonoscopy can detect *T. trichiura* in challenging or severe cases, but biopsies are required to confirm a diagnosis [44]. Ultrasonography may have a role in diagnosing hepatobiliary and pancreatic ascariasis as well as trichiuriasis, showing the whipworm dance on ultrasound, and this is a modality that can easily be used in resource-poor settings [45]. In trichiuris dysentery syndrome, an evaluation of IDA is essential [7].

Having a high index of suspicion is the key for diagnosing strongyloidiasis. The diagnosis can only be confirmed when the worm is identified in the stool. Microscopic-based techniques have a low sensitivity and are strongly dependent on the technician’s expertise. To perform the Baermann technique or agar plate culture, a series of analyses of multiple specimens taken over 3 days is necessary, as it is often impossible to detect the worm from a single stool sample due to both low worm burden and intermittent release of larvae [46]. Serology for *S. stercoralis* does not answer the question of whether the infection is recent or from the past and cured, so this technique could overestimate the prevalence of the disease. White blood cell count is important in simple and uncomplicated infections, when eosinophilia is a helpful symptom. However, it is not constant and substantially absent in disseminated strongyloidiasis [47]. Molecular methods to diagnose *S. stercoralis* infection have yet to be improved [47].

## 8. Treatment

Classic hookworm disease can be managed with anthelmintic treatment and iron therapy that may be supported by an appropriate diet. Iron supplementation; additional nutritional support (including folate supplementation); blood transfusion, usually followed by a diuretic, and the monitorization of treatment should be considered in patients with severe disease [7].

Currently, the benzimidazole anthelmintic drugs mebendazole and albendazole are the treatment of choice for hookworms, *A. duodenale*, and *T. trichiura*. 

Benzimidazoles have excellent safety profiles in doses used to treat hookworm infection, although transient abdominal pain, diarrhoea, nausea, dizziness, and headache might occur [7,48]. Albendazole administered at 400 mg once daily for three days, mebendazole administered at 500 mg once, or mebendazole administered at 100 mg twice daily for three days is the treatment of choice for hookworms. In experimental animal studies, these drugs have been shown to be teratogenic and thus are not recommended in the first trimester of pregnancy. To date, teratogenicity in humans has not been observed [49]. A systematic review by Gyorkos and colleagues highlighted that none of the studies reporting on adverse birth outcomes that they included found a statistically or clinically significant difference in the frequency of adverse events between the group exposed to albendazole or mebendazole, and the unexposed group [50]. The safety of benzimidazoles has not been established for children younger than 12 months [51]. 

Mebendazole and albendazole are the treatment of choice for *T. Trichiura*: 3 days of 400 mg of albendazole once daily, 500 mg of mebendazole once daily, or 100 mg of mebendazole twice daily. Either ivermectin (200 mcg/kg daily) or pyrantel embonate (11 mg/kg once daily for 3 days) is an alternative option. Patients with severe or symptomatic anaemia should be considered for iron supplementation, and supportive treatment is strongly recommended in the case of dysentery [7]. The effectiveness of the treatment should be monitored, as anthelmintic drugs are only partially effective for *T. trichiura* infections.

For an *A. lumbricoides* infection, 400 mg of albendanzole or 500 mg of mebendazole in a single oral dose, or 100 mg of mebendazole twice daily for 3 days is recommended in patients older than 12 months who do not require urgent surgical approach. Ivermectin in a single dose of 200 mcg/kg daily is an alternative option [7,47]. Laparotomy might be needed in cases of obstruction, volvulus, and intussusception, and anthelmintic treatment can be given when the patient’s conditions are stable [33]. 

Ivermectin is the first-line treatment for strongyloidiasis [7], based on a Cochrane meta-analysis, which demonstrated that ivermectin has a better safety profile than thiabendazole, which shows similar efficacy to ivermectin and superior efficacy to albendazole [52]. A recent randomized trial showed that multiple doses of ivermectin did not have a higher efficacy and was also less tolerated than a single dose (200 µg/kg). Thus, a single dose should be recommended for the treatment of non-disseminated strongyloidiasis [53]. For STH infections, eosinophil counts should be repeated at least one month after the end of treatment, and the patient’s reversion into the normal range should be monitored. Seven to fourteen days after the end of treatment, laboratory follow-up based on egg counts is recommended for all STH infections. A serology analysis after 6–12 months plus agar plate culture or Bearmann technique may be performed for follow-up of strongyloidiasis [54].

## 9. Conclusions

Anaemia has multifactorial causes, including infectious agents. Among the latter, helminths are often neglected as determinants of or as a contributing factor to anaemia. With this review, we aim to raise awareness of the causative role of helminthic agents, particularly for health personnel operating in non-endemic countries.

## Data Availability

Not applicable.

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
