# Peer review of "Soil-Transmitted Helminths and Anaemia: A Neglected Association Outside the Tropics"

_microorganisms, 2022, doi:10.3390/microorganisms10051027_

Round 1

Author Response

Dear Reviewer,

We have carefully read all the comments received and would like to thank you for all your suggestions and constructive comments, which allow us to improve our manuscript.

All questions were developed in the manuscript. Here I would like to work out the question about the interactions between inflammation and coagulation, as suggested in point 7: “Under A. duodenale mechanisms, the text suggests that anticoagulants downregulate the immune response. This seems counterintuitive and should be cited”

The inflammation process initiates clotting, decreases the activity of natural anticoagulant mechanisms and impairs the fibrinolytic system. Inflammatory cytokines are the major mediators involved in coagulation activation. Some components of the natural anticoagulant cascades, like thrombomodulin, minimize endothelial cell dysfunction by rendering the cells less responsive to inflammatory mediators, facilitate the neutralization of some inflammatory mediators and decrease loss of endothelial barrier function. Hence hookworms not only reduces thrombosis but also reduces the inflammatory process by promoting the anticoagulant pathways. As acknowledged, parasites need to be able to exploit the host without damaging it excessively, so all these mechanisms are focused on maintaining an immunotolerance necessary for the survival of the parasite itself.

This strategy is also used in the clinical practice for the prevent the  hemostatic imbalance in sepsis, characterized by the excessive activation of procoagulant pathways and the impairment of anticoagulant activity, leads to disseminated intravascular coagulation and results in microvascular thrombosis and tissue hypoperfusion. In this cases, a supportive strategies aiming at inhibiting activation of coagulation and inflammation by treatment with exogenous anticoagulants which are in clinical use (antithrombin and activated protein C), have been found to be beneficial.

About this:

Esmon CT. The interactions between inflammation and coagulation. Br J Haematol. 2005 Nov;131(4):417-30. doi: 10.1111/j.1365-2141.2005.05753.x. PMID: 16281932.

Esmon CT. The impact of the inflammatory response on coagulation. Thromb Res. 2004;114(5-6):321-7. doi: 10.1016/j.thromres.2004.06.028. PMID: 15507261.

Feistritzer C, Wiedermann CJ. Effects of anticoagulant strategies on activation of inflammation and coagulation. Expert Opin Biol Ther. 2007 Jun;7(6):855-70. doi: 10.1517/14712598.7.6.855. PMID: 17555371.

Reviewer 2 Report

The manuscript entitled “Soil-transmitted helminths and anemia: a neglected association outside the tropics” is generally well-written and offers succinct information on several helminth species that are known to infect humans, and notably, their important yet often neglected role in causing anaemia disease. There are a couple of places with very minor English errors or extra spaces between words, but these can all be easily fixed later by MDPI English editing team. It was a definitely good read and one that is fit for publication in Microorganisms.

Author Response

The reviewer 2 recommended that the manuscript should corrected for very minor English errors.

As suggested, the text has been checked using a English editing support.